# Do mothers pick up a phone? A cross-sectional study on delivery of MCH voice messages in Lagos, Nigeria

Kazuya Ogawa[1,2]*, Yoshito Kawakatsu[3,4], Nobuhiro Kadoi[5], Olukunmi Omobolanle Balogun[6], Adefunke Oyeniyi Adesina[7], Veronica Olubunmi Iwayemi[8], Hirotsugu Aiga[5,9]

1 Department of Ecoepidemiology, Institute of Tropical Medicine (NEKKEN), Nagasaki University, Nagasaki, Japan, 2 Graduate School of Arts and Letters, Tohoku University, Sendai, Japan, 3 Department of Global Health, University of Washington, Seattle, WA, United States of America, 4 Graduate School of Biomedical Sciences, Nagasaki University, Nagasaki, Japan, 5 Human Development Department, Japan International Cooperation Agency (JICA), Tokyo, Japan, 6 National Center for Child Health and Development, Tokyo, Japan, 7 Lagos State Ministry of Health / Nigeria Feild Epidemiology and Laboratory Training Program, Feltp, Nairobi, Nigeria, 8 Lagos State Primary Health Care Board (LSPHCB), Lagos, Nigeria, 9 School of Tropical Medicine and Global Health, Nagasaki University, Nagasaki, Japan

* ogawa.kazuya.p5@dc.tohoku.ac.jp

## Abstract

### Background

Voice messages have been employed as an effective and efficient approach for increasing health service utilization and health promotion in low- and middle-income countries. However, unlike SMS, voice message services require their users to pick up a phone call at its delivery time. Furthermore, voice messages are difficult for the users to review their contents afterward. While recognizing that voice messages are more friendly to specific groups (eg, illiterate or less literate populations), there should be several challenges in successfully operationalizing its intervention program.

### Objective

This study is aimed to estimate the extent to which voice message service users pick up the phone calls of voice messages and complete listening up to or beyond the core part of voice messages.

### Methods

A voice message service program composed of 14 episodes on maternal, newborn, and child health was piloted in Lagos, Nigeria, from 2018 to 2019. A voice message call of each of 14 episodes was delivered to the mobile phones of the program participants per day for 14 consecutive days. A total of 513 participants in the voice message service chose one of five locally spoken languages as the language to be used for voice messages. Two multi-level logistic regression models were created to understand participants' adherence to the voice message: (a) Model 1 for testing whether a voice message call is picked up; and (b)

the authors for researchers who obtain the approval from the Ministry of Health, Lagos State (lagresunit@gmail.com), and the relevant Ethical Review Committee (dcst@lasuth.org).

**Funding:** This work was supported by Japan International Cooperation Agency (JICA). No specific grant ID number was assigned. The funders had no role in study design, data collection and analysis, decision to publish, or preparation of the manuscript.

**Competing interests:** The authors declare no competing interests.

**Abbreviations:** IVR, Interactive Voice Response; JICA, Japan International Cooperation Agency; LGA, Local Government Area; LSPHCB, Lagos State Primary Health Care Board; SMS, Short Message Service.

Model 2 for testing whether a voice message call having been picked up is listened to up to the core messaging part.

## Results

The greater the voice message episode number became, the smaller proportion of the participants picked up the phone calls of voice message (aOR: 0.98; 95% CI: 0.97–0.99; P = .01). Only 854 of 3765 voice message calls having been picked up by the participants (22.7%) were listened to up to their core message parts. It was found that picking up a phone call did not necessarily ensure listening up to the core message part. This indicates a discontinuity between these two actions.

## Conclusions

The participants were likely to stop picking up the phone as the episode number of voice messages progressed. In view of the discontinuity between picking up a phone call and listening up to the core message part, we should not assume that those picking up the phone would automatically complete listening to the entire or core voice message.

## Introduction

The World Health Organization (WHO) defines mHealth as the use of mobile and wireless technologies to support health objectives [1]. Interventions using mHealth include short messaging service (SMS), application, and voice message [1]. Voice message is recommended as an intervention method in a WHO guideline [2].

In developing countries, voice message has been used for HIV [3–5] and diabetes patients [6] to improve and maintain their medication adherence, and for commercial sex workers (CSWs) to get screened for sexually transmitted infections [7]. In the area of maternal and child health (MCH) care, voice message has been used to encourage less literate [8] and less educated [9] mothers to increase their use of health services. In addition, voice message has helped low-income pregnant women to take iron tablets [10] and parents to improve infant and young child feeding practices [11]. In a systematic review, voice message was reviewed as the second commonest mHealth intervention in low-income countries, followed by short message service (SMS) [12].

In Nigeria, where this study was conducted, there are several reported cases of mHealth programs. SMS was employed to improve medication adherence in outpatients [13] and malaria patients [14]. In the area of MCH, SMS was used to increase uptake of vaccinations [15] and postnatal care services (PNC) [14]. Free mobile phone call services were used to increase mothers' visits to medical facilities and use of their services [16]. It was reported that voice message interventions have helped increase screening for fistula [17] and promote the use of contraceptives [18].

Several previous studies confirmed the effects of voice message interventions. For example, a study in Bangladesh found that voice messages increased the knowledge of newborn health among pregnant women [19]. Another study in India found that voice messages increased tetanus toxoid vaccine uptake and facility-based deliveries among pregnant women [20]. A randomized controlled trial in India found that voice messages delivered to pregnant women contributed to weight gain among newborns [21].

It has been reported that voice messaging interventions are positively accepted [3, 5, 8, 9, 22–24]. However, a voice message intervention requires its participants to pick up the phone at the time of delivery, unlike SMS intervention [25]. This feature suggests that the voice message intervention may not always be delivered as intended. Thus, in some cases, the participants could not pick up the phone and listen to voice message. For example, a previous study in Afghanistan reported that the participants having opted for voice messages ended up receiving fewer messages than those having opted for SMS [9]. In an intervention for pregnant women in Ghana, more than 80% of the participants picked up the phone and listened to at least 50% of the length of each message prenatally, while less than 54% of the participants did so postnatally [26]. The aforementioned study in Bangladesh reported that only a third of the participants (29.9%) picked up six or more phone calls of a total of eight voice messages delivered in a month. The primary reason for not picking up the phone was the commitment to household chores. Additionally, only 35.5% of those having picked up a phone completed listening to voice messages to the end [19]. In India, most participants (88.1%) picked up the phone and completed listening to a 19-second voice message targeting CSWs. However, only 58.5% of the participants completed listening to a 31-second voice message [7]. These studies imply that ensuring participants' adherence to voice messages is crucial for successful voice message intervention. Therefore, understanding adherence to the voice message needs to be deepened to achieve the expected effect of this intervention.

In this study targeting the area in Lagos that includes urban slum communities, 14 consecutive voice messages on MCH were delivered to the participants. In the urban slum communities, mothers were less likely to use MCH services, and the maternal mortality ratio was high (1050 per 100000 live births), compared to that of Lagos State (545 per 100000 live births) [27]. Having assumed that only half of the women were literate in the area [27] and most the women in Lagos State (85.9%) owned mobile phones [28], voice message on MCH was employed as an appropriate intervention.

The participants' adherence to a voice message was divided into two phases: (a) picking up a phone call; and (b) listening at least up to the core message part of a voice message. Then, we attempted to assess the intervention adherence in picking up the phone calls and in completing listening up to the core message part, in accordance with the progress of a series of maternal and child health voice messages.

## Methods

This study is a cross-sectional study using two multilevel logistic regression models.

### Study site

This study was conducted in Lagos Mainland, Lagos State, whose population was estimated at 13.5 million as of 2018 [29]. The study site was Lagos Mainland Local Government Area (LGA), one of the most populous LGAs in Lagos State, Nigeria [30]. While Yoruba is the largest ethnic group in Lagos State, there are other ethnic groups such as Egun, Hausa and Igbo [31]. According to the Nigeria Demographic Health Survey 2018, most women (85.9%) in Lagos State owned mobile phones [28].

The study site was one of the targets LGAs of the Project for Strengthening Pro-Poor Community Health Services in Lagos State (the Project), implemented jointly by Lagos State Primary Health Care Board (LSPHCB) and Japan International Cooperation Agency (JICA) during the period from January 2017 to March 2019. The Project aimed to strengthen the primary healthcare service delivery system for urban poor populations. To evaluate the project

interventions, the Project conducted a baseline survey in February 2017 and its follow-up survey in July 2018.

### Recruitment of study participants

The study target groups were pregnant women having participated in both the Project's baseline and follow-up surveys and further expressed their willingness to receive voice messaging interventions (Fig 1). In the follow-up survey, we randomly selected 1000 from the cohort of

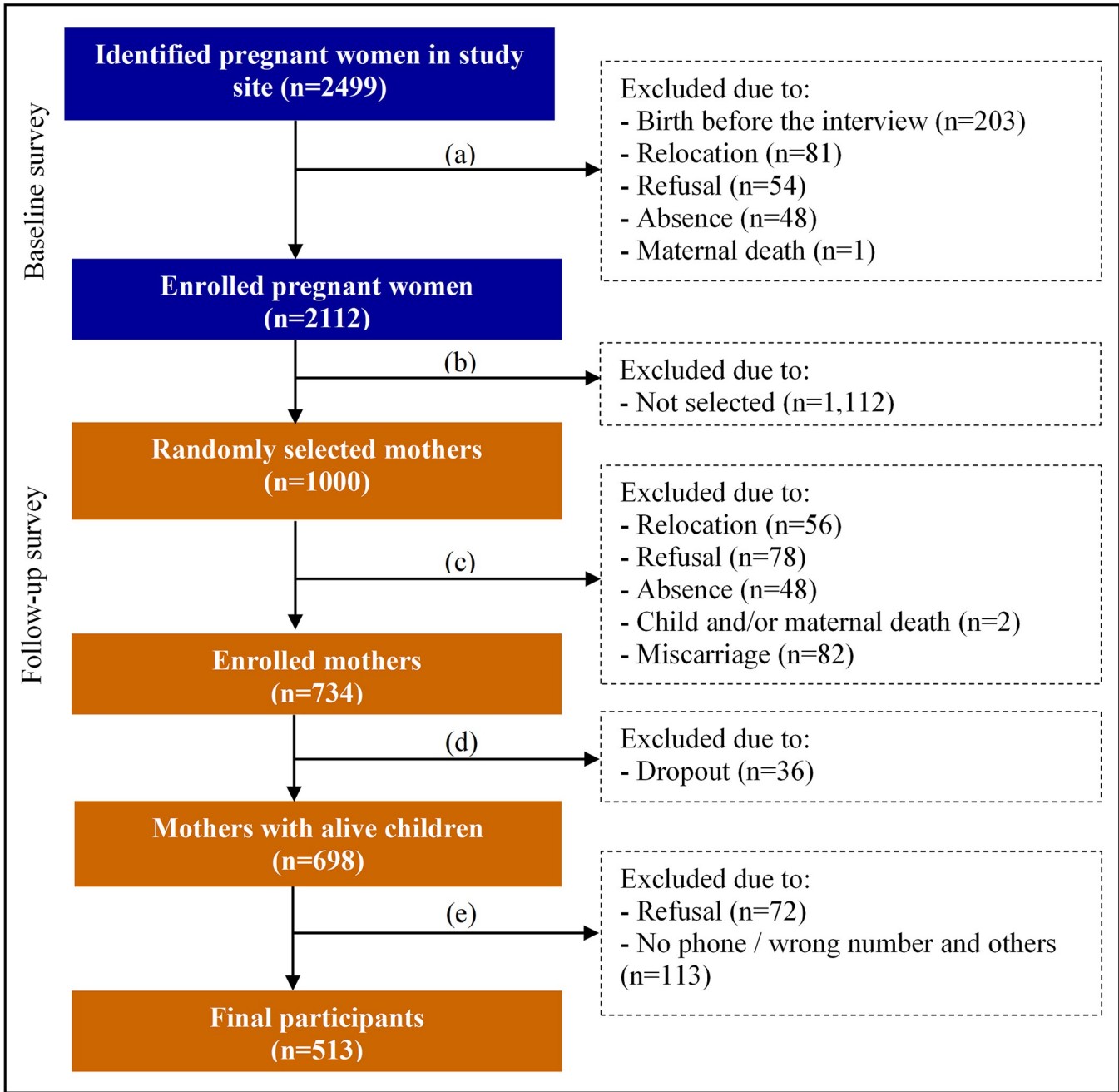

**Fig 1. Flow of the sampling procedure.**

2112 pregnant women who participated in the baseline survey conducted in 2017. The number of pregnant women in the baseline survey (n = 2112) was great enough to represent entire pregnant women in the study site. Of 1000 pregnant women selected from the cohort defined in the baseline survey, 698 mothers having given live births were interviewed in the follow-up survey. The reasons for not participating in the follow-up survey included: relocation (n = 56), refusal (n = 78), absence (n = 48), child and/or maternal death (n = 2) and miscarriage (n = 82) ((c) in Fig 1). Of 698 interviewed mothers, 513 indicated the ownership and availability of mobile phones in their households and agreed to receive voice messages as an intervention of this study.

## Voice messaging intervention

The voice messaging intervention consisted of 14 episodes in different topics: (a) introduction; (b); antenatal care; (c) danger signs during pregnancy; (d) postnatal care; (e) newborn care; (f) child illnesses #1; (g) child illnesses #2; (h) exclusive breastfeeding; (i) immunization; (j) complementary feeding; (k) vitamin A supplementation; (l) growth monitoring; (m) prevention of child accidents; and (n) conclusion (Table 1). The messages were composed and then reviewed several times jointly by the health education experts of Lagos State Ministry of Health (LSMOH), LSPHCB, and JICA. The messages were initially composed in English and then translated into five local languages: Egun, Hausa, Igbo, Pidgin English, and Yoruba. Participants received voice messages in one of the five local languages they had chosen in advance. Each episode lasted for approximately 170 seconds, starting with an opening music, a topic-specific dialogue between an announcer and a nurse, and then ending with a closing music. Participants received one episode per day for 14 consecutive dayson their mobile phones registered during the baseline survey. We randomly assigned the participants to either of the two groups. One group received voice messages for 14 days from 8 to 21 December 2018 (Group

**Table 1. Structure of the voice message program.**

| Episode number | Topic of episode | Mean length of voice message of five local language versions (sec) | Time threshold | |
|---|---|---|---|---|
| | | | Mean length of core message part of five local language versions (sec) | Mean proportion of length of core message part to that of entire voice message (%) |
| 1 | Introduction | 163.4 | 81.2 | 49.7 |
| 2 | Antenatal care | 168.4 | 102.6 | 60.9 |
| 3 | Danger signs during pregnancy | 158.2 | 105.4 | 66.6 |
| 4 | Postnatal care | 177.4 | 130.8 | 73.7 |
| 5 | Newborn care | 168.2 | 110.8 | 65.9 |
| 6 | Child illnesses #1 | 168.6 | 109.0 | 64.7 |
| 7 | Child illnesses #2 | 176.4 | 121.4 | 68.8 |
| 8 | Exclusive breastfeeding | 168.8 | 107.8 | 63.9 |
| 9 | Immunization | 165.4 | 113.0 | 68.3 |
| 10 | Complementary feeding | 177.6 | 151.2 | 85.1 |
| 11 | Vitamin A supplementation | 160.0 | 104.8 | 65.5 |
| 12 | Growth monitoring | 163.2 | 99.6 | 61.0 |
| 13 | Prevention of child accidents | 174.0 | 129.4 | 74.4 |
| 14 | Conclusion | 162.8 | 124.8 | 76.7 |
| Overall mean | | 168.0 | 113.7 | 67.7 |

1), and the other group did for 14 days from 7 to 20 January 2019 (Group 2). During the first period of message delivery (ie, for Group 1), the same message was broadcasted at least once a day through three local radio stations: (a) Radio Lagos for Yoruba and Egun; (b) Eko FM for Hausa, Igbo and Pidgin English; and (c) Traffick FM for Pidgin English. During the second period of the message delivery (ie, for Group 2), no radio broadcast was made. We planned to randomly allocate the voice message delivery time to each participant from the following three options: (a) initial call at 10 AM and reminder call at noon; (b) initial call at noon and reminder at 2 PM; and (c) initial call at 10 AM and reminder call at 2 PM. Only when a participant did not pick up the initial phone call, another call was delivered as the reminder either at noon or 2 PM. However, a certain proportion of voice messages (12.8%) were delivered after 4 PM, most likely due to technical errors of the voice message system.

## Data source

This study was designed as a cross-sectional study using the following two types of datasets. First, to identify the characteristics of the participants, we used the baseline and follow-up survey data collected by the Project. Those surveys collected the socio-demographic and socio-economic status data using an interviewer-administered structured questionnaire on a computer-assisted personal interview (CAPI) software SurveyCTO (ver 2.20, Dobility Inc., Massachusetts). Second, we used the output data from the voice message system that was operated and managed by a local system development company (eg, participants' phone numbers, languages participants chose, the episode numbers of voice messages, dates and time of voice message delivery, and the number of minutes during which participants listened to each voice message). When a participant picked up a phone call, the voice message system automatically recorded its time and date, and length of listening time. When a participant failed to pick up an initial phone call, she then had one more chance to receive the voice message delivery as aforementioned. When a participant did not pick up the reminder call, the voice message system recorded the message delivery time and response status (ie, busy, ringing but unanswered).

## Regression models

In this study, two multilevel logistic regression models were developed. Model 1 tested whether a voice message call was picked up (whether participants started listening to a voice message), while Model 2 tested whether a voice message call having been picked up was listened to up to the core messaging part. Thus, Model 2 was applied exclusively to those having picked up voice message calls.

## Dependent variables

A dichotomous variable, whether a participant picked up a voice message call (ie, "Picked" and "Did not pick"), was employed as the dependent variable for Model 1. "Picked" was coded when the participant picked up the phone. "Did not pick" was coded when the call record was either busy or ringing but unanswered in the voice message system. A dichotomous variable, whether a participant completed listening up to the core message part (ie, "Completed" and "Did not complete"), was employed as the dependent variable for Model 2.

The minimum number of seconds for which a voice message needs to be listened to for participants' adequate and meaningful understanding was set as the time threshold for each episode (Table 1). When a participant hung up the phone without completing listening up to the time threshold, we assumed that the level of her understanding on the message was partial and inadequate. Thus, "Completed" indicates that a participant listened to the message up to or

beyond the time threshold. Alternatively, "Did not complete" indicates that a participant hung up before the time threshold.

### Independent variables

A total of 14 variables were employed as independent variables for both Model 1 and Model 2. They were composed of: (a) six variables related to mothers (age, religion, marital status, education, employment status, and language chosen for receiving voice messages); (b) two variables related to children born to them during the intervention period (age and sex); (c) three variables related to households (the number of children under five years of age, decision-maker on health, and wealth quintile); and (d) three variables related to the voice messaging intervention (implementation year and month, ownership of mobile phone, episode number of a voice message, and voice message delivery time).

Age of children born to participant mothers during the intervention period was classified into three categories as the date of birth was unknown for some children: (a) under 18 months of age; (b) 18–24 months of age; and (c) NA. Wealth quintile was created by sorting out all the mothers' households according to the wealth index values. Wealth index was calculated by applying a principal components analysis [32] to variables of households' ownerships of key properties and access to key services (water source, sanitation facility, cooking fuel, materials for floor, roof and external walls, radio, television, refrigerator, generator, fan, air conditioner, computer, bicycle, motorbike, car). All the mothers' households were divided into five equal-sized groups by wealth index score (ie, poorest, poor, middle, rich, and richest). A series of voice message episodes were numbered from 1 to 14 according to the order of maternal and child health milestones and message deliveries.

Most of the independent variables are shown in Table 2, but only voice message delivery time is shown in Table 3. The independent variables are presented separately in these two tables because the total number of cases differs between participant-related variables (n = 513 in Table 2) and intervention-result-related variables (n = 7182 in Table 3). Message delivery time was classified into four categories: (a) between 10 AM and noon; (b) between noon and 2 PM; (c) between 2 PM and 4 PM; and (d) after 4 PM.

### Data analysis

The dependent variables for both Model 1 and Model 2 are dichotomous. Since 14 voice messages were delivered to all the participants, the data on participants' adherence to each 14 voice message were recorded in the voice message system. This study employed multilevel logistic regression analysis with random effect and reported an adjusted odds ratio at 95% confidence interval (CI) and $P$ value using the robust standard error for each model. All data processing and analyses were performed using Stata (ver 15.1, StataCorp LLC, College Station, TX).

### Ethical consideration

Ethical approval was obtained from the Health Research and Ethics Committee at Lagos State University Teaching Hospital (Ref: LREC /06/10/764). Written informed consent was obtained from all participants.

## Results

### Characteristics of participants

As shown in Table 2, mean age of the participants was 28.5 years (SD 5.5). More than half of the mothers (338/513, 65.9%) were Christian, and most (480/513, 93.6%) were married. Those

**Table 2. Socio-demographic and socio-economic characteristics of study participants (n = 513).**

| Socio-demographic and socio-economic characteristics | | Value |
|---|---|---|
| **Mother's age (years), mean (SD)** | | 28.5 (5.5) |
| **Mother's religion, n (%)** | | |
| | Christian | 338 (65.9) |
| | Non-Christian | 175 (34.1) |
| **Mother's marital status, n (%)** | | |
| | Married | 480 (93.6) |
| | Never married, divorced and widowed | 33 (6.4) |
| **Mother's education attainment, n (%)** | | |
| | Never went to school and preschool | 75 (14.6) |
| | Primary education | 129 (25.1) |
| | Secondary education | 227 (44.3) |
| | Tertiary education | 82 (16.0) |
| **Mother's employment status, n (%)** | | |
| | Unemployed and homemaker | 144 (28.1) |
| | Self-employment | 329 (64.1) |
| | Full-time and temporary employment | 40 (7.8) |
| **Language chosen for receiving voice messages, n (%)** | | |
| | Yoruba | 246 (48.0) |
| | Pidgin English | 188 (36.6) |
| | Others (Egun, Hausa, Igbo) | 79 (15.4) |
| **Age of the youngest child (month), n (%)** | | |
| | <18 months of age | 85 (16.6) |
| | 18–24 months of age | 412 (80.3) |
| | NA | 16 (3.1) |
| **Sex of the youngest child, n (%)** | | |
| | Male | 278 (54.2) |
| | Female | 235 (45.8) |
| **Number of children under five years old in household, mean (SD)** | | 1.5 (0.7) |
| **Decision-maker on health in household, n (%)** | | |
| | Husband or partner | 339 (66.1) |
| | Herself | 128 (25.0) |
| | Husband and herself (joint decision) | 39 (7.6) |
| | Parents or other | 7 (1.3) |
| **Wealth quintile, n (%)** | | 0.3 (1.0) |
| | Richest | 106 (20.6) |
| | Richer | 104 (20.3) |
| | Middle | 102 (19.9) |
| | Poorer | 101 (19.7) |
| | Poorest | 100 (19.5) |
| **Implementation year and month, n (%)** | | |
| | Group 1: 8 – 21 December 2018 | 261 (50.1) |
| | Group 2: 7 – 20 January 2019 | 252 (49.0) |
| **Ownership of mobile phone, n (%)** | | |
| | Herself | 420 (81.9) |
| | Not herself | 93 (18.1) |

**Table 3. Voice messaging intervention characteristics (n = 7182).**

| Voice messaging intervention results | | Value |
|---|---|---|
| **Total number of participants** | | 513 |
| **Total number of voice messages per participant** [a] | | 14 |
| **Total number of voice messages delivered from voice message system** | | 7182 |
| **Number of voice messages not recorded in voice message system** | | 32 |
| **Number of voice messages not having reached participants due to network error** | | 751 |
| **Voice message delivery time, n (%)** | | |
| | 10 AM–noon | 1704 (26.6) |
| | Noon– 2 PM | 2199 (34.4) |
| | 2 PM– 4 PM | 1675 (26.2) |
| | 4 PM– | 821 (12.8) |
| **Total number of voice messages analyzed for Model 1** | | 6399 |
| **Model 1 (Picked up a phone call), n (%)** | | |
| | Picked | 3765(58.8) |
| | Did not pick | 2634 (41.2) |
| **Total number of voice messages analyzed for Model 2** | | 3765 |
| **Model 2 (Completed listening up to or beyond time threshold), n (%)** | | |
| | Completed listening | 854 (22.7) |
| | Did not complete listening | 2911 (77.3) |

[a] Exclude reminding messages

having completed secondary or tertiary education accounted for more than half of the participants (309/513, 60.3%). Only a sixth (75/513, 14.6%) of the participants did not complete formal education. The most common type of employment was self-employment (329/513, 64.1%). The languages most frequently chosen for the interview by participants were Yoruba, followed by Pidgin English, Egun, Hausa and Igbo. The proportion of males was slightly greater than females among children born to the participants during the intervention (278/513, 54.2%). Most of them (412/513, 80.3%) were aged between 1.5 and 2 years. Mean number of children under five years of age per household was 1.5 (SD 0.7). Husbands or other types of partners were more likely to make health-related decisions in the households (339/513, 66.1%). A great majority of the participants (420/513, 81.9%) in the study had their mobile phones.

## Voice message delivery

Table 3 shows the summary of the voice messaging intervention results. One voice message was sent to all the 513 participants per day for 14 consecutive days. The total number of voice messages delivered was 7182. Of them, almost all the messages (7150/7182, 99.6%) were successfully recorded in the voice message system, while the rest (32/7182, 0.4%) were not recorded probably for technical reasons of the system. Of 7150 delivered messages, 751 did not successfully reach participants' phone numbers due to network errors. Therefore, they were excluded from further data analysis. As a result, Model 1 was applied to 6399 voice message cases. Approximately one-third of voice messages (2199/6399, 34.4%) were delivered to participants' phone numbers between noon and 2 PM.

Of the 6399 voice messages, less than half (2634/6399, 41.2%) were not responded to by participants. Thus, only a little more than half (3765/6399, 58.8%) were responded. Of the 3765 messages, one-fifth (854/3765, 22.7%) were listened to up to or beyond the time threshold. Of

**Table 4. Participants' adherence to voice messages.**

| Episode cluster | Proportion of voice massage call having been picked up (%) | Proportion of voice massage calls having completed being listened up to or beyond time threshold when being picked up (%) | Mean length up to time threshold among five local languages (sec) | Mean call length when being picked up (sec) |
|---|---|---|---|---|
| Episode cluster 1 (Episode #1-#5) | 61.3% (1407/2296) | 28.4% (400/1407) | 106.2 | 67.8 |
| Episode cluster 2 (Episode #6-#10) | 57.9% (1314/2269) | 19.1% (252/1314) | 120.5 | 68.0 |
| Episode cluster 3 (Episode #11-#14) | 56.9% (1044/1834) | 19.3% (202/1044) | 114.7 | 67.1 |

the 513 participants, 50 (9.7%) never responded to any of the 14 voice messages delivered (incl. reminder messages), while 42 (8.2%) responded to all voice messages. Of all 14 voice message calls, 6.5 were picked up on average.

Table 4 and Fig 2 show the proportions of participants who picked up the phone and listened to voice messages up to or beyond the time thresholds. The greater the episode number became, the lower the proportion of participants completed listening up to core contents. Thus, as the voice message episode number progressed, fewer mothers completed listening up

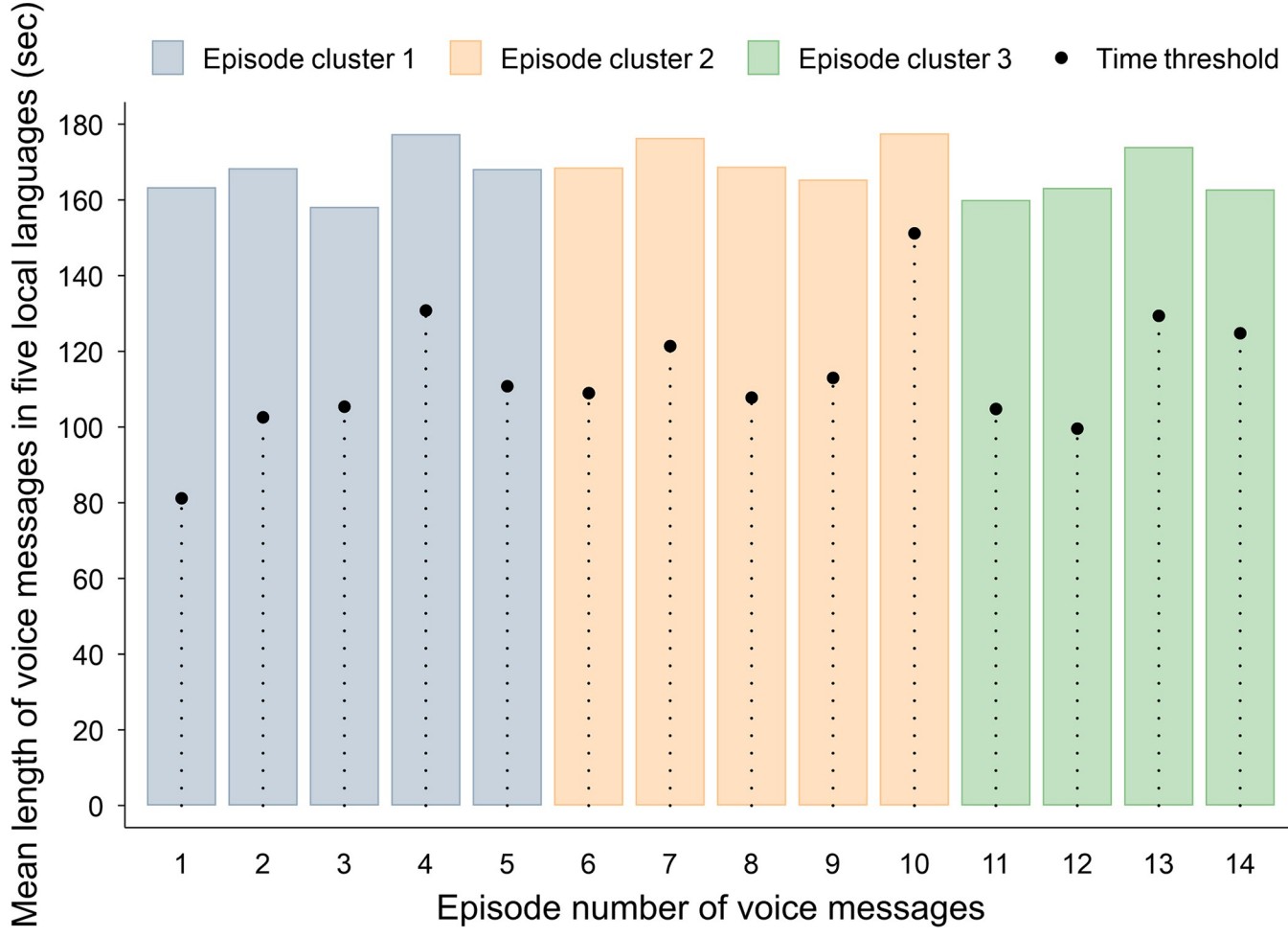

**Fig 2. Mean length of voice messages and time threshold in five local languages by episode number.**

to core contents. The proportion of voice message calls having been picked up by the participant was 61.3% (1407/2296) for the first five episodes, 57.9% (1314/2269) for the following five episodes, and 56.9% (1044/1834) for the last four episodes.

Furthermore, the number of voice message calls having been picked but hung up before the time threshold increased in accordance with the progress of the episode numbers. The proportion of voice message calls having completed being listened up to or beyond the time threshold was 28.4% (400/1407) for the initial five episodes (#1-#5 in Table 4), 19.1% (252/1314) for the following five episodes (#6-#10 in Table 4), and 19.3% (202/1044) for the last four episodes (#11-#14 in Table 4). However, there was no significant difference in mean number of seconds for which participants listened to voice messages between 14 voice messages. Instead, the time threshold of the voice message was likely to become longer as the episode number (order of voice messages) increased. This implies that the higher proportion of participants stopped listening before the time threshold was attributed not to less time spent listening but the greater number of seconds to the time threshold.

## Regression analyses

**Model 1: Whether to pick up a phone call.**   As shown in Table 5, of the 15 independent variables for Model 1 regression, three (language chosen for receiving voice messages, message delivery time, and episode number of voice messages) were significantly associated with responses to a phone call (ie, busy, ringing but unanswered).

Voice message calls in Pidgin English were 0.75 times less likely to be picked up than those in Yoruba (aOR: 0.75; 95% CI: 0.61–0.94; $P$ = .01). Compared with the voice messages delivered between noon and 2 PM, those delivered between 2 PM and 4 PM were 0.82 times less likely to be picked up (aOR: 0.82; 95% CI: 0.69–0.97; $P$ = .02). The same trend was confirmed in those delivered after 4 PM (aOR: 0.79; 95% CI: 0.63–0.99; $P$ = .04).

As the episode number progressed day by day during 14 consecutive days, the voice message calls became 0.98 times less likely to be picked up (ie, per day) (aOR: 0.98; 95% CI: 0.97–0.99; $P$ = .01).

**Model 2: Whether to complete listening up to core message part.**   As shown in Table 5, of the 14 independent variables for Model 2 regression, three (ie, language chosen for receiving voice messages, delivery time, and episode number) were significantly associated with picking up a phone.

Voice message calls in Pidgin English were 1.32 times more likely to complete being listened up to or beyond the time threshold than those in Yorba (aOR: 1.32; 95% CI: 1.03–1.69; $P$ = .03). Compared with voice messages delivered between noon and 2 PM, those delivered between 10 AM and noon were 0.76 times less likely to complete listening to up to the time threshold (aOR: 0.76; 95% CI: 0.60–0.97; $P$ = .03). The greater the episode number became, the smaller the likelihood that voice messages were listened to up to the time threshold would be (aOR: 0.93; 95% CI: 0.91–0.95; $P$ < .001).

**Sensitivity analysis.**   We conducted sensitivity analyses using two types of samples. The first type was the samples excluding the first and last episodes (eg, introduction and conclusion), while the second type was the samples excluding mothers not owning mobile phones. Of the 14 voice message episodes, the first and final ones did not address specific topics related to maternal and child health (eg, ANC and PNC) but rather served as introduction and conclusion sessions, respectively. Therefore, these two-episode numbers were excluded from the analysis. It was confirmed that the results did not change with this sub-dataset.

Additionally, this study included 18.1% (93/513) of cases where the mothers did not own mobile phone(s) even when there was a mobile phone in the household. Under this situation,

**Table 5. Logistic regression models for picking up a phone call (Model 1) and listening to a voice message up to or beyond time threshold (Model 2).**

| Independent variable | | Model 1: Whether a phone call was picked | | Model 2: Whether a voice message was listened to up to or beyond time threshold when being picked up | |
|---|---|---|---|---|---|
| | | Adjusted OR (95% CI) | *P* value | Adjusted OR (95% CI) | *P* value |
| **Mother's age (years)** | | 1.00 (0.99–1.02) | 0.670 | 1.01 (0.99–1.03) | 0.396 |
| **Mother's religion** | | | | | |
| | Christian | Ref. | | Ref. | |
| | Non-Christian | 1.03 (0.83–1.28) | 0.786 | 1.13 (0.86–1.48) | 0.388 |
| **Marital status** | | | | | |
| | Married | Ref. | | Ref. | |
| | Never married, divorced, and widowed | 1.10 (0.73–1.68) | 0.645 | 1.05 (0.66–1.68) | 0.831 |
| **Mother's education attainment** | | | | | |
| | Never went to school and preschool | 0.81 (0.55–1.20) | 0291 | 1.26 (0.87–1.83) | 0.224 |
| | Primary education | 0.97 (0.77–1.24) | 0.824 | 1.02 (0.75–1.38) | 0.922 |
| | Secondary education | Ref. | | Ref. | |
| | Tertiary and higher education | 1.13 (0.83–1.53) | 0.454 | 0.71 (0.48–1.04) | 0.082 |
| **Mother's employment status** | | | | | |
| | Unemployed and homemaker | 1.02 (0.83–1.26) | 0.812 | 1.18 (0.92–1.52) | 0.193 |
| | Self-employment | Ref. | | Ref. | |
| | Full-time and temporary employment | 1.09 (0.79–1.49) | 0.609 | 1.38 (0.90–2.11) | 0.138 |
| **Language chosen for receiving voice messages** | | | | | |
| | Yoruba | Ref. | | Ref. | |
| | Pidgin English | 0.75 (0.61–0.94) | 0.011 | 1.32 (1.03–1.69) | 0.031 |
| | Others (Egun, Housa, Igbo) | 1.06 (0.78–1.45) | 0.677 | 1.42 (0.91–2.20) | 0.121 |
| **Age of the youngest child (month)** | | | | | |
| | <18 month of age | 0.81 (0.63–1.04) | 0.094 | 0.84 (0.60–1.18) | 0.313 |
| | 19–24 month of age | Ref. | | Ref. | |
| | NA | 0.81 (0.44–1.47) | 0.493 | 1.06 (0.50–2.24) | 0.876 |
| **Number of children under five years of age in household** | | 1.06 (0.94–1.21) | 0.330 | 1.01 (0.87–1.19) | 0.860 |
| **Sex of the youngest child** | | | | | |
| | Male | Ref. | | Ref. | |
| | Female | 1.01 (0.85–1.22) | 0.878 | 1.14 (0.90–1.44) | 0.282 |
| **Decision maker on health in household** | | | | | |
| | Husband and Partner | Ref. | | Ref. | |
| | Herself | 1.23 (0.99–1.55) | 0.066 | 1.07 (0.81–1.42) | 0.637 |
| | Husband and Herself | 1.27 (0.88–1.82) | 0.196 | 1.29 (0.82–2.01) | 0.271 |
| | Parents and Other | 1.18 (0.57–2.46) | 0.644 | 0.53 (0.17–1.61) | 0.260 |
| **Wealth quintile** | | | | | |
| | Richest | 0.89 (0.59–1.34) | 0.571 | 1.17 (0.72–1.88) | 0.528 |
| | Richer | 0.70 (0.49–1.01) | 0.056 | 0.95 (0.62–1.46) | 0.825 |
| | Middle | 0.83 (0.58–1.18) | 0.298 | 1.30 (0.86–1.96) | 0.216 |
| | Poorer | 0.93 (0.68–1.27) | 0.662 | 1.04 (0.72–1.49) | 0.829 |
| | Poorest | Ref. | | Ref. | |
| **Implementation year and month** | | | | | |
| | December 2018 | Ref. | | Ref. | |
| | January 2019 | 1.03 (0.85–1.23) | 0.774 | 1.13 (0.90–1.43) | 0.280 |
| **Voice message delivery time** | | | | | |
| | 10 AM–noon | 1.07 (0.91–1.25) | 0.435 | 0.76 (0.60–0.97) | 0.025 |
| | Noon– 2 PM | Ref. | | Ref. | |

(*Continued*)

**Table 5.** (Continued)

| Independent variable | | Model 1: Whether a phone call was picked | | Model 2: Whether a voice message was listened to up to or beyond time threshold when being picked up | |
|---|---|---|---|---|---|
| | | Adjusted OR (95% CI) | *P* value | Adjusted OR (95% CI) | *P* value |
| | 2 PM– 4 PM | 0.82 (0.69–0.97) | 0.019 | 1.04 (0.82–1.31) | 0.767 |
| | 4 PM– | 0.79 (0.63–0.99) | 0.042 | 0.74 (0.54–1.01) | 0.060 |
| Ownership of mobile phone | | | | | |
| | Herself | Ref. | | Ref. | |
| | Not herself | 0.91 (0.70–1.19) | 0.502 | 1.00 (0.74–1.36) | 0.984 |
| Episode number | | 0.98 (0.97–0.99) | 0.006 | 0.93 (0.91–0.95) | <0.001 |

it was largely possible that someone in a household other than the mother picked up the phone and further listened to voice messages. Therefore, we conducted another sensitivity analysis using samples limited to the cases where mobile phone owners were mothers. The results did not change except for voice message delivery time, which became statistically insignificant for Model 1 and Model 2 with this restriction.

## Discussion

### Picking up a phone of voice messages

Each of the 14 voice message calls (170 seconds on average) was delivered once a day consecutively for 14 days. Only 61.3% (1407/2296) of voice message calls were picked up for the initial part of 14 episodes (eg, Episode #1- #5). Furthermore, Model 1 regression analysis results indicated that participants were likely to stop picking up the phone as the episode number progressed [25, 26].

Two types of challenges concerning the voice message intervention are herewith raised. One of the possible reasons for not picking up a phone is that a voice message call is, unlike SMS, required to be picked up exclusively at the time of its delivery [25]. The participants were granted maximum two opportunities per day to receive a voice message. Maximum two opportunities might not have been sufficient for the participants. This is because most of them are urban poor women whose daily movements are generally more frequent than a middle-income group [33]. More than half of them (329/513, 64.1%) work by engaging themselves in informal economic activities where maternity and parental leave systems do not exist. Even those with infants might neither necessarily stay at home nor be ready to pick up a phone during the daytime. Second, the proportion of voice massage calls having been picked up was reduced over time as the episode number progressed. In a previous study in Ghana, more than 80% of women listened to the voice messages during pregnancy, but only 54% listened after delivery [26]. The result of this previous study was consistent with that of our study on the point that the probability of picking up/listening to the voice message calls decreased over time. It is assumed that participants lost their interest in the voice messages as the episode number progressed. However, this reason needs to be confirmed through qualitative research such as focus group discussions.

### Completing listening to voice messages

In a previous study, the time threshold (the minimum number or proportion of seconds for which a voice message needs to be listened to for meaningful understanding) was uniformly set at 50% of the entire voice message [26]. This study set an episode-specific time threshold by

carefully reviewing each voice message(eg, 49.7% for Episode #1, 60.9% for Episode #2, and 66.6% for Episode #3). Thus, having employed a more precisely defined time threshold, this study should be able to present more meaningful results of analyses.

Model 2 regression analysis results indicate that even those having picked up a phone became less likely to complete listening up to and beyond the time threshold as the episode number progressed. The proportion of voice message calls having completed being listened to up to or beyond the time threshold was 28.4% (400/1407) for the initial five episodes (Episode #1- #5), 19.1% (252/1314) for the subsequent five episodes (Episode #6 - #10), and 19.3% (202/1044) for the final four episodes (Episode #11- #14). Mean length to the time threshold tended to become longer toward Episode #14, ie, 106.2 seconds for the initial five episodes (Episode #1- #5), 120.5 for the subsequent five episodes (Episode #6 - #10), and 114.7 for the final four episodes (Episode #11 - #14), respectively. On the other hand, mean length of time during which participants listened to a voice message did not change significantly across the episodes, ie, 67.8 seconds for the first five episodes (Episode #1- #5), 68.0 seconds for the following five episodes (Episode #6 - #10), and 67.1 seconds for the last four episodes (Episode #11 - #14), respectively. That is why, as a result, the proportion of participants having completed listening up to or beyond the time threshold became smaller as voice message episode progressed. Therefore, we simply cannot conclude that the participants in this study spent less time listening to the voice messages over time.

Nevertheless, note that voice messages completed listening to up to or beyond the time thresholds were only a fifth (854/3765, 22.7%) among those picked up. This result suggests that picking up a phone does not necessarily ensure listening to the core message part. A previous study also reported that only a third of participants (35.5%) picked up a phone and completed listening to all the messages [19]. Therefore, health policymakers and planners should be warned that picking up a phone does not necessarily lead to the completion of listening to a voice message.

## Length of voice messages

The voice messages in this study took the form of a conversation between an announcer and a nurse to ensure client-friendliness by capturing the mothers' attention. As a result, mean length of each voice message was 170 seconds, ranging from 130 to 190 seconds.

A study on voice messaging intervention targeting pregnant women of urban poor households in India reported that all those having picked up a phone completed listening to the entire voice message that lasted for 40 seconds [10]. Another study in India similarly reported that 93% of participants completed listening to the entire voice message, which was 19 seconds long [7]. Our study found that mothers listened to a voice message for 68 seconds on average once they picked up a phone. Hence, compared with the results of other studies, the voice messages in this study were too long to complete listening. Only half of the voice messages (2056/3765, 54.6%) were listened to for more than 60 seconds by those having picked up a phone. The appropriate length of the voice message might have been 60 seconds at the longest.

## Voice message delivery timing

The initial voice message calls were delivered randomly either at 10 AM or noon. Those not having picked up a phone received its reminder call either at noon or 2 PM. Several previous studies reported that participants received the voice messages at the time predesignated by themselves [9, 10, 26, 34].

As indicated in Model 1 regression analysis, participants to whom voice messages were delivered at 2 PM tended to be less likely to pick up a phone than those to whom voice

messages were delivered at noon. Voice messages delivered at noon included both the initial and reminder calls, while voice messages delivered at 2 PM were exclusive reminder calls for those not having picked up the initial calls. This implies that those not having picked up the initial calls were also less likely to pick up the reminder calls. They were probably less interested in voice messages.

Unlike an SMS, the contents of a voice message cannot be reviewed later but listened to only when it is delivered [25]. If they cannot take time for some reason when a voice message is delivered, they will either not pick up the phone or hang up the phone immediately. As shown in Model 2 regression analysis results, the participants having picked up the phone between 10 AM and noon were less likely to complete listening up to or beyond the time threshold (core message part) than those having picked up the phone between noon and 2 PM. This may imply that some participants were too busy with economic activities or household chores [9, 19] to pick up a phone in the morning. In fact, most participants (369/513, 71.9%) were either employed or self-employed. Thus, digital health planners may have to pay more careful attention to message delivery timing when designing a voice message intervention. However, as indicated in the results of sensitivity analyses, the difference in the proportion of those having picked up the phones by delivery timing was insignificant when excluding the mothers who did not own mobile phones. Therefore, our findings from this study on delivery time may be inconclusive, and further research is needed.

## Languages chosen for receiving voice messages

The participants were requested to choose a language to be used for voice messages from four: Yoruba, Pidgin English, and other languages (Egun, Hausa and Igbo were combined into one group for analyses). Most participants (434/513, 84.6%) chose either Yoruba or Pidgin English. This language preference distribution among the participants was consistent with the previous study on linguistics in Nigeria [31]. Furthermore, another previous study using IVR found no association between ethnicity and the rate of IVR completion, although it is the ethnicity, not the language [34].

Model 1 regression analysis results show that participants having chosen Pidgin English were less likely to pick up the phone than those having chosen Yoruba. Alternatively, Model 2 regression analysis results indicate that those having chosen Pidgin English were more likely to complete listening up to or beyond the time threshold than those having chosen Yorba.

Probably, this is not because the time threshold for Pidgin English voice messages tends to be longer than that for Yoruba. Rather, overall mean length of time up to the time threshold for 14 episodes was nearly equal (122.3 seconds in Pidgin English and 120.4 seconds in Yoruba). The proportion of voice message calls that were picked up and further listened to up to or beyond the time threshold was 23.8% (315/1322) for Pidgin English and 20.9% (391/1871) for Yoruba. This implies that participants having chosen Pidgin English were more likely to complete listening up to the core message part than those having chosen Yoruba once they picked up the phone, despite the same level of the time threshold between Pidgin English and Yoruba. Nevertheless, the participants having chosen Pidgin English were less likely to pick up the phone than those having chosen Yoruba.

## Parallel radio broadcasting of voice messages

We randomly divided participants into two groups and delivered voice messages in 8–21 December 2018 (Group 1) and 7–20 January 2019 (Group 2). When Group 1 participants received 14 voice messages in December 2018, the same contents as voice messages were broadcasted by a local radio station. The objective of broadcasting the same message was to

encourage them to continue and complete listening to voice messages, by familiarizing participants with the music and messages to be delivered to their phones. This radio program was not implemented when Group 2 participants received 14 messages in January 2019. As Model 1 and Model 2 regression analyses show, there was no statistically significant difference between Group 1 and Group 2 in the proportion of participants picking up a phone and further listening up to or beyond the time threshold.

There might be a possibility that part of those having listened to the contents of a voice message on the radio no longer listened to the same contents on their mobile phone again. However, the results suggest that the radio broadcasting (the same message in parallel) had at least no negative impact on picking up the phone and listening to voice messages. Unfortunately, the extent to which the participants listened to the radio message is unknown. Therefore, further study is needed to develop an evidence-based strategy for the synergetic implementation of voice messaging on the phone and social messaging through media.

### Implications for policymakers

Voice message intervention requires its participants to pick up the phone at the time of delivery and listen up to the core contents of message, unlike SMS intervention. Therefore, it is crucial to ensure participants' adherence to voice messages to achieve the expected effects of this intervention. The results of this study imply that the following adjustments need to be made to improve the adherence to voice message intervention: (a) limiting the total number of episodes and shortening the voice message program duration to let participants be focused (ie, reduce from 14 episodes in this study); (b) shortening the length of each voice message to enable participants to complete listening (ie, reduce from 170 seconds on average in this study); and (c) deliver more than two reminder calls (ie, two reminder calls in this study) for those unable to pick up the phone at the time of delivery.

### Limitations of the study

We implemented this voice messaging intervention in an area that includes urban slum communities in Nigeria, where mobile phone ownership and acceptance of mHealth had been assumed to be high enough. However, 10.3% (72/698) of intervention target individuals did not desire to participate in the intervention (Fig 1). This attrition may affect the generalizability of the study results.

A certain proportion of voice messages (821/6399, 12.8%) were not delivered in time as initially scheduled (eg, after 4 PM). This type of delivery error could be attributed to technical problems of the voice message system such as: (a) delivery delay of voice messages initially scheduled at 2 PM; and (b) the third delivery of voice messages to those not having picked up the initial and reminder calls. Such unscheduled message delivery was reported in a previous study as well [26]. The results of this study may not fully capture the association between delivery timing of voice messages and whether participants pick up the phones and complete listening to voice messages.

### Conclusion and recommendations

The participants were likely to stop picking up a phone as the episode number of voice message progressed. Furthermore, picking up a phone did not necessarily ensure listening up to the core message part. Thus, health policymakers and planners should not take it for granted that local individuals continue to pick up the phone throughout the program and listen to the messages once they are delivered. To enable target populations to pick up the phone and complete listening up to or beyond the core message part of voice message, each voice message

should be concise, and the total number of episodes should also be smaller. Moreover, multiple reminder calls should be necessary. To deepen the understanding on the reasons for participants to stop picking up the phone, qualitative studies need to be conducted.

## Supporting information

**S1 Questionnaire. Inclusivity in global research.**
(DOCX)

## Acknowledgments

We would like to thank Lagos State Ministry of Health and Lagos State Primary Health Care Board for their technical and logistic supports. Special thanks go to all the mothers having participated in this study.

## Author Contributions

**Conceptualization:** Yoshito Kawakatsu, Nobuhiro Kadoi, Hirotsugu Aiga.

**Data curation:** Yoshito Kawakatsu.

**Formal analysis:** Kazuya Ogawa, Olukunmi Omobolanle Balogun.

**Investigation:** Yoshito Kawakatsu, Nobuhiro Kadoi, Adefunke Oyeniyi Adesina, Veronica Olubunmi Iwayemi, Hirotsugu Aiga.

**Methodology:** Kazuya Ogawa, Yoshito Kawakatsu, Nobuhiro Kadoi, Hirotsugu Aiga.

**Project administration:** Nobuhiro Kadoi.

**Software:** Kazuya Ogawa.

**Supervision:** Hirotsugu Aiga.

**Validation:** Kazuya Ogawa.

**Visualization:** Kazuya Ogawa.

**Writing – original draft:** Kazuya Ogawa.

**Writing – review & editing:** Kazuya Ogawa, Yoshito Kawakatsu, Nobuhiro Kadoi, Olukunmi Omobolanle Balogun, Adefunke Oyeniyi Adesina, Veronica Olubunmi Iwayemi, Hirotsugu Aiga.

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
