## [Decision Letter · Decision Letter 0]

30 Mar 2022

PONE-D-22-00657Do mothers pick up a phone? A cross-sectional study on utilizations of MCH voice messages in Lagos, NigeriaPLOS ONE

Dear Dr. Ogawa,

Thank you for submitting your manuscript to PLOS ONE. After careful consideration, we feel that it has merit but does not fully meet PLOS ONE’s publication criteria as it currently stands. Therefore, we invite you to submit a revised version of the manuscript that addresses the points raised during the review process.

The reviewers are largely positive on this paper, but have made several suggestions for framing, organization, and discussion that I agree will improve it.==============================

We look forward to receiving your revised manuscript.

Kind regards,

Emily W. Harville

Academic Editor

PLOS ONE

Journal Requirements:

“This work was supported by Japan International Cooperation Agency (JICA). No specific grant ID number was assigned.”

Please note that funding information should not appear in the Funding section or other areas of your manuscript. We will only publish funding information present in the Funding Statement section of the online submission form.

“This work was supported by Japan International Cooperation Agency (JICA). No specific grant ID number was assigned. The funders had no role in study design, data collection and analysis, decision to publish, or preparation of the manuscript.”

Reviewers' comments:

Reviewer's Responses to Questions

**Comments to the Author**

1. Is the manuscript technically sound, and do the data support the conclusions?

Reviewer #1: Yes

Reviewer #2: Yes

2. Has the statistical analysis been performed appropriately and rigorously? 

Reviewer #1: Yes

Reviewer #2: Yes

3. Have the authors made all data underlying the findings in their manuscript fully available?

Reviewer #1: No

Reviewer #2: Yes

4. Is the manuscript presented in an intelligible fashion and written in standard English?

Reviewer #1: No

Reviewer #2: Yes

5. Review Comments to the Author

Reviewer #1: Title

a) I don’t think the title reflects the focus of the study, particularly in terms of utilization of the voice messages. Utilization, in the context of this study, would mean that participants heard the messages and utilized them in some way in their daily life. This study seems to be more about delivery of the intervention.

Introduction

a) Please explain why this particular voice messaging intervention was implemented in Nigeria with this population.

b) The phrasing of the study purpose at the top of page 5 is a bit confusing: “to assess the changes in the participants’ behavior in picking up the phone calls and in completing listening to the core message part.” The study seems to be about intervention adherence, not behavior change. For example, was the intervention successfully delivered as intended? Picking up the phone and actually listening to the core content of the message is part of adherence. I might suggest re-framing or re-writing the study purpose and supplementing the introduction with a paragraph on why intervention adherence is important.

Results

a) Sensitivity analysis section – It’s not clear what is meant by “We confirmed significant differences in the aforementioned variables were robust.”

b) Sensitivity analysis section – It’s not clear what is meant by “The significant difference in the proportion of those having picked up the phones by message delivery timing were not robust (become insignificant) for both Models, when excluding mothers not owning mobile phones.”

Discussion

a) Results are repeated quite a bit in the discussion section.

b) What does this mean? “This result suggests that picking up a phone and listening up to the core message part were sequential with a low probability, rather implying a discontinuity between the two actions.” Do you mean that picking up the phone does not necessarily mean they will listen to the message?

c) On page 24 this statement does not appear to be supported by the data. Reasons for not picking up a phone were not assessed, nor were perceptions of the messages as either interesting or useful. “Not surprisingly, those having thought voice messages are neither interesting nor useful enough at the initial stage of a series of voice message episodes would tend increasingly not to pick up a phone.”

d) What do the results imply for intervention adherence?

English language – It would be important to have the manuscript thoroughly edited for word choice, phrasing, style, verb tense, grammar etc. As a native English speaker, it was difficult for me to understand certain statements.

Reviewer #2: In general, this is a nice piece of work. But there is some few point that can be improved by adding the following point below:

a) You can structure the introduction as the first paragraph on worldwide, the second paragraph on low and middle income countries, the third paragraph on the systematic reviews. The last paragraph on single studies, and study done in Nigeria on mHealth program.

b) Voice call is part of the mHealth program, I did not see any information in this regard. Please provide some information on mHealth and MCH.

c) On page 5 you stated that this study targeting the area that includes urban slum communities in Lagos, Nigeria, the participants’ response to a voice message was divided into two phases. But you did not give your reader the actual mHealth program situation in Nigeria. It will be crucial to provide some literature about the mHealth for MCH in Nigeria. Such the intervention modality (mobile phone call, messages and son on…)

d) On results section, please be consistent in way that you are presenting the percentage

6. PLOS authors have the option to publish the peer review history of their article (what does this mean?). If published, this will include your full peer review and any attached files.

Reviewer #1: No

Reviewer #2: No

---

## [Author Response · Author response to Decision Letter 0]

26 May 2022

Reviewer #1: Title　

a) I don’t think the title reflects the focus of the study, particularly in terms of utilization of the voice messages. Utilization, in the context of this study, would mean that participants heard the messages and utilized them in some way in their daily life. This study seems to be more about delivery of the intervention.

Response: 

We appreciate and agree with the reviewer’s comment. The title has been corrected as follows. 

Do mothers pick up a phone?

A cross-sectional study on delivery of MCH voice messages in Lagos, Nigeria

Introduction

a) Please explain why this particular voice messaging intervention was implemented in Nigeria with this population.

Response: 

We thank the reviewer for the reviewer’s instructive suggestion. Following this, we have added why this intervention was carried out with this population in the 6th paragraph of introduction (P5).

In this study targeting the area in Lagos that includes urban slum communities, 14 consecutive voice messages on MCH were delivered to the participants. In the urban slum communities, mothers were less likely to use MCH services, and the maternal mortality ratio was high (1050 per 100000 live births), compared to that of Lagos State (545 per 100000 live births)[27]. Having assumed that only half of the women were literate in the area[27] and most women in Lagos State (85.9%) owned mobile phones[28], voice message on MCH was employed as an appropriate intervention.

b) The phrasing of the study purpose at the top of page 5 is a bit confusing: “to assess the changes in the participants’ behavior in picking up the phone calls and in completing listening to the core message part.” The study seems to be about intervention adherence, not behavior change. For example, was the intervention successfully delivered as intended? Picking up the phone and actually listening to the core content of the message is part of adherence. I might suggest re-framing or re-writing the study purpose and supplementing the introduction with a paragraph on why intervention adherence is important.

Response: 

We agree with the reviewer in that picking up the phone and listening to the core content of the message is part of adherence.　Following this suggestion, the authors have added a paragraph on why intervention adherence is important and revised the study purpose as follows.

A paragraph added in the 5th paragraph of introduction (P5):

These studies imply that ensuring participants' adherence to voice messages is crucial for successful voice message intervention. Therefore, understanding adherence to the voice message needs to be deepened to achieve the expected effect of this intervention.

A sentence revised in the 7th paragraph of introduction (P6): 

Then, we attempted to assess the intervention adherence in picking up the phone calls and in completing listening up to the core message part, in accordance with the progress of a series of maternal and child health voice messages.

Results

a) Sensitivity analysis section – It’s not clear what is meant by “We confirmed significant differences in the aforementioned variables were robust.”

Response: 

We thank the reviewer. This has been corrected as follows (P23). 

It was confirmed that the results did not change with this sub-dataset.

b) Sensitivity analysis section – It’s not clear what is meant by “The significant difference in the proportion of those having picked up the phones by message delivery timing were not robust (become insignificant) for both Models, when excluding mothers not owning mobile phones.”

Response: 

We thank the reviewer. This has been corrected as follows (P23). 

The results did not change except for voice message delivery time, which became statistically insignificant for Model 1 and Model 2 with this restriction.

Discussion

a) Results are repeated quite a bit in the discussion section.

Response:

We appreciate and agree with the reviewer’s comment. The authors removed the sentences below.

A sentence removed from 1st paragraph of discussion (P24):

Those having failed to pick up the initial call for a voice message subsequently received its reminder call after at least two hours on the same day.

A sentence removed from 1st paragraph of discussion (P24):

First, less than half of overall voice message calls (2634/6399, 41.2%) were not picked up.

Sentences removed from 1st paragraph of discussion (P24):

The participants in this study received the initial voice message randomly either at 10 AM or at noon. Those not having picked up the initial voice message call received a reminder call either at noon or at 2 PM.

b) What does this mean? “This result suggests that picking up a phone and listening up to the core message part were sequential with a low probability, rather implying a discontinuity between the two actions.” Do you mean that picking up the phone does not necessarily mean they will listen to the message?

Response: 

We appreciate the reviewer’s comment. The meaning of this sentence is the same as the reviewer has indicated. The following is the modified sentence (P26).

This result suggests that picking up a phone does not necessarily ensure listening to the core message part.

c) On page 24 this statement does not appear to be supported by the data. Reasons for not picking up a phone were not assessed, nor were perceptions of the messages as either interesting or useful. “Not surprisingly, those having thought voice messages are neither interesting nor useful enough at the initial stage of a series of voice message episodes would tend increasingly not to pick up a phone.”

Response:

We agree with the reviewer in that this sentence is not supported by data. This has been modified as follows (P24).

In a previous study in Ghana, more than 80% of women listened to the voice messages during pregnancy, but only 54% listened after delivery[26]. The result of this previous study was consistent with that of our study on the point that the probability of picking up/listening to the voice message calls decreased over time. It is assumed that participants lost their interest in the voice messages as the episode number progressed. However, this reason needs to be confirmed through qualitative research such as focus group discussions.

d) What do the results imply for intervention adherence?

Response:

We thank the reviewer for the reviewer’s insightful comment. Following this comment, the authors have added a paragraph on implication for intervention adherence as follows (P31).

Voice message intervention requires its participants to pick up the phone at the time of delivery and listen up to the core contents of message, unlike SMS intervention. Therefore, it is crucial to ensure participants' adherence to voice messages to achieve the expected effects of this intervention. The results of this study imply that the following adjustments need to be made to improve the adherence to voice message intervention: (a) limiting the total number of episodes and shortening the voice message program duration to let participants be focused (ie, reduce from14 episodes in this study); (b) shortening the length of each voice message to enable participants to complete listening (ie, reduce from 170 seconds on average in this study); and (c) deliver more than two reminder calls (ie, two reminder calls in this study) for those unable to pick up the phone at the time of delivery.

English language – It would be important to have the manuscript thoroughly edited for word choice, phrasing, style, verb tense, grammar etc. As a native English speaker, it was difficult for me to understand certain statements.

We thank the reviewer for pointing the mistakes. The authors have reviewed and revised the entire manuscript.

Reviewer #2: In general, this is a nice piece of work. But there is some few point that can be improved by adding the following point below:

a) You can structure the introduction as the first paragraph on worldwide, the second paragraph on low and middle income countries, the third paragraph on the systematic reviews. The last paragraph on single studies, and study done in Nigeria on mHealth program.

Response:

We appreciate and agree with the reviewer’s comment. Following this comment, we have modified the structure of the Introduction (P3-6).

b) Voice call is part of the mHealth program, I did not see any information in this regard. Please provide some information on mHealth and MCH.

Response:

We appreciate the reviewer’s comment. The authors have added the information on mHealth and MCH as follows.

P3 

Interventions using mHealth include short messaging service (SMS), application, and voice message[1].

P3

In the area of maternal and child health (MCH) care, voice message has been used to encourage less literate[8] and less educated[9] mothers to increase their use of health services.

P3

In the area of MCH, SMS was used to increase uptake of vaccinations[15] and postnatal care services (PNC)[14]. Free mobile phone call services were used to increase mothers' visits to medical facilities and use of their services[16].

c) On page 5 you stated that this study targeting the area that includes urban slum communities in Lagos, Nigeria, the participants’ response to a voice message was divided into two phases. But you did not give your reader the actual mHealth program situation in Nigeria. It will be crucial to provide some literature about the mHealth for MCH in Nigeria. Such the intervention modality (mobile phone call, messages and son on…)

Response:

We thank the reviewer for the reviewer’s insightful comment. Following this comment, the authors have added a paragraph about the mHealth for MCH in Nigeria as follows (P3).

In Nigeria, where this study was conducted, there are several reported cases of mHealth programs. SMS was employed to improve medication adherence in outpatients[13] and malaria patients[14]. In the area of MCH, SMS was used to increase uptake of vaccinations[15] and postnatal care services (PNC)[14]. Free mobile phone call services were used to increase mothers' visits to medical facilities and use of their services[16]. It was reported that voice message interventions have helped increase screening for fistula[17] and promote the use of contraceptives[18].

d) On results section, please be consistent in way that you are presenting the percentage

Response:

We thank the reviewer. This has been corrected by following this comment.

---

## [Decision Letter · Decision Letter 1]

7 Sep 2022

PONE-D-22-00657R1Do mothers pick up a phone? A cross-sectional study on delivery of MCH voice messages in Lagos, NigeriaPLOS ONE

Dear Dr. Ogawa,

Thank you for submitting your manuscript to PLOS ONE. After careful consideration, we feel that it has merit but does not fully meet PLOS ONE’s publication criteria as it currently stands. Therefore, we invite you to submit a revised version of the manuscript that addresses the points raised during the review process.

The reviewers are in agreement that the major points have been addressed, but have added a couple of minor points to be addressed to finalize the paper.==============================

We look forward to receiving your revised manuscript.

Kind regards,

Emily W. Harville

Academic Editor

PLOS ONE

Journal Requirements:

Reviewers' comments:

Reviewer's Responses to Questions

**Comments to the Author**

1. If the authors have adequately addressed your comments raised in a previous round of review and you feel that this manuscript is now acceptable for publication, you may indicate that here to bypass the “Comments to the Author” section, enter your conflict of interest statement in the “Confidential to Editor” section, and submit your "Accept" recommendation.

Reviewer #1: All comments have been addressed

Reviewer #3: All comments have been addressed

2. Is the manuscript technically sound, and do the data support the conclusions?

Reviewer #1: Yes

Reviewer #3: Yes

3. Has the statistical analysis been performed appropriately and rigorously? 

Reviewer #1: Yes

Reviewer #3: Yes

4. Have the authors made all data underlying the findings in their manuscript fully available?

Reviewer #1: Yes

Reviewer #3: No

5. Is the manuscript presented in an intelligible fashion and written in standard English?

Reviewer #1: Yes

Reviewer #3: Yes

6. Review Comments to the Author

Reviewer #1: The authors have fully addressed all of the comments in my review. I recommend that it now be accepted for publication.

Reviewer #3: (No Response)

7. PLOS authors have the option to publish the peer review history of their article (what does this mean?). If published, this will include your full peer review and any attached files.

Reviewer #1: No

Reviewer #3: No

---

## [Author Response · Author response to Decision Letter 1]

19 Sep 2022

We appreciate the editor’s comment. The reference list has been corrected as the attached document (Response to Reviewers.docx).

---

## [Editor Report · Decision Letter 2]

26 Sep 2022

Do mothers pick up a phone? A cross-sectional study on delivery of MCH voice messages in Lagos, Nigeria

PONE-D-22-00657R2

Dear Dr. Ogawa,

We’re pleased to inform you that your manuscript has been judged scientifically suitable for publication and will be formally accepted for publication once it meets all outstanding technical requirements.

Kind regards,

Emily W. Harville

Academic Editor

PLOS ONE